# Effect of Graft and Nano ZnO on Nutraceutical and Mineral Content in Bell Pepper

**DOI:** 10.3390/plants10122793

**Published:** 2021-12-17

**Authors:** José-Gerardo Uresti-Porras, Marcelino Cabrera-De-La Fuente, Adalberto Benavides-Mendoza, Emilio Olivares-Sáenz, Raul I. Cabrera, Antonio Juárez-Maldonado

**Affiliations:** 1Doctorado en Ciencias en Agricultura Protegida, Universidad Autónoma Agraria Antonio Narro, Saltillo 25315, Mexico; 2Departamento de Horticultura, Universidad Autónoma Agraria Antonio Narro, Saltillo 25315, Mexico; cafum7@yahoo.com (M.C.-D.-L.F.); abenmen@gmail.com (A.B.-M.); 3Facultad de Agronomía, Universidad Autónoma de Nuevo León, Francisco I. Madero S/N, Ex Hacienda el Canada, General Escobedo 66050, Mexico; emolivares@gmail.com; 4Rutgers Agricultural Research and Extension Center (RAREC), Department of Plant Biology, Rutgers University, Bridgeton, NJ 08302, USA; cabrera@njaes.rutgers.edu; 5Departamento de Botánica, Universidad Autónoma Agraria Antonio Narro, Saltillo 25315, Mexico; antonio.juarez@uaaan.edu.mx

**Keywords:** biostimulation, induced stress, nutritional value, fruit quality, healthy

## Abstract

The objective of this experiment was to evaluate the effects of grafting, zinc oxide nanoparticles (ZnO NPs), and their interaction on the nutritional composition of bell pepper plants. The treatments evaluated included grafted and non-grafted pepper plants with four concentrations of ZnO NPs (0, 10, 20, 30 mg L^−1^) applied to the foliage. The following parameters were evaluated: content of N, P, K^+^, Ca^2+^, Mg^2+^, Mn^2+^, Zn^2+^, Fe^2+^, Cu^2+^, total antioxidants, ascorbic acid, total phenols, glutathione, total proteins, fruit firmness, and total soluble solids. Grafting increased the content of N 12.2%, P 15.9%, K^+^ 26.7%, Mg^2+^ 20.3%, Mn^2+^ 34.7%, Zn^2+^ 19.5%, Fe^2+^ 18.2%, Cu^2+^ 11.5%, antioxidant capacity 2.44%, ascorbic acid 4.63%, total phenols 1.33%, glutathione 7.18%, total proteins 1.08%, fruit firmness 8.8%. The application of 30 mg L^−1^ ZnO NPs increased the content of N 12.3%, P 25.9%, Mg^2+^ 36.8%, Mn^2+^ 42.2%, Zn^2+^ 27%, Fe^2+^ 45%, antioxidant activity 13.95%, ascorbic acid 26.77%, total phenols 10.93%, glutathione 11.46%, total proteins 11.01%, and fruit firmness 17.7% compared to the control. The results obtained demonstrate the influence of the use of grafts and ZnO NPs as tools that could improve the quality and nutrient content in fruits of bell pepper crops.

## 1. Introduction

Bell peppers contain a wide range of nutritional and bio-functional properties related to their phenolic compounds, flavonoids, capsaicinoids, carotenoids, tocopherols, vitamins C, A, E, B, potassium, magnesium, iron, calcium, and phosphorus [1,2]. In 2019, the world production of bell pepper was 1,990,926 hectares, of which a fruit production of 38,027,164 tons was obtained, with an average yield of 19.10 tons per hectare [3]. Mexico reported 152,772.55 hectares planted in 2019, with 3,238,244.81 tons harvested, with an average yield of 21.65 tons per hectare [4].

Zinc is very important in the human diet, crucial for gene expression, needed for the activity of several metalloenzymes [5], essential for various cellular processes such as differentiation, apoptosis, and proliferation, DNA metabolism and repair, reproduction, and vision, all of which influence growth and development of an organism [6]. Human Zn^2+^ deficiency affects approximately two billion people worldwide, especially children below 5 years and pregnant women [7]. Biofortification appears more practical, and it can be performed at farms to produce Zn-enriched cereal grains for combating widespread deficiency in developing countries [8]. Biofortification can be implemented through agronomic management (e.g., soil/foliar application of fertilizers) [9].

Zinc application improves yield and yield components through various mechanisms. For example, it improves chlorophyll content and triggers photosynthetic activity and auxin synthesis, which lead to better growth and development of the crop [10]. Zinc deficiency in plants affects root development, and therefore the absorption of water and plant available nutrients from the soil [11]. It is also crucial for many physiological processes, including enzyme activation, protein synthesis, and nucleic acid and carbohydrate metabolism [12]. Zinc nanoparticles applied to plants through foliar application (aerosol spray) can be absorbed efficiently and translocated to the different plant parts [13]. Zinc toxicity depends on different factors: plant species, dose, application form, application route, and type of soil [14].

Cultural practices such as grafting provide benefits such as greater vigor, plant development, and defense against some stresses [15,16]. Grafting is a special plant propagation technique, which consists of joining an aerial part of the plant (scion) joining another root part (rootstock), where both grow together to form a new plant [17]. The appropriate rootstock can manipulate the morphology of the scion and help deal better with stresses [18,19], such as improving the uptake, transport, and use of nutrient ions [20].

The rapid development of nanotechnology has facilitated the transformations of the traditional food and agricultural sectors, in particular the invention of smart and active packaging, nanosensors, nanopesticides, and nanofertilizers [21]. A nanoparticle is defined as any particle designed with a dimension of 1 to 100 nanometers and has properties that are not shared by non-nanoscale particles with the same chemical composition [22,23]. The use of nanoparticle technology applied to crops can enhance the penetration of ions through the stomata [24], then translocated by the vascular bundles of the xylem and phloem to other tissues [25,26].

Organisms need metallic elements as nutrients to facilitate their metabolism and other biological activities. NPs significantly influence various physiological aspects and increase root growth and yield of horticultural crops [27]. ZnO NPs, in particular, have the advantage of high activity, stability, and effectiveness in nutrient delivery compared to their solid or ionic counterparts [28,29]. The nanometric size of NPs covers larger surface areas that could more efficiently absorb, translocate, and retain nutrients in plants [30]. Zinc is one of the essential nutrients required for plant growth. Its important role can be adjudged as it controls the synthesis of indole acetic acid, a phytohormone that dramatically regulates plant growth [31].

However, little is known about the effect that ZnO NPs have on the mineral and nutraceutical composition of grafted bell pepper plants cultivated in intensively managed hydroponic systems such as Nutrient Film Technique NFT. Therefore, the objective of this work was to evaluate the effect of grafting and foliarly applied ZnO NPs on the nutritional value of bell pepper. The leading hypothesis in this work was that the nutritional content of fruits is positively modified by grafting and the use of ZnO NPs in pepper crops.

## 2. Results

### 2.1. Graft Effect on Mineral Contents

Grafting favored the assimilation of mineral elements in the pepper plants, reflecting a significant difference between them except for Ca^2+^ content (Table 1). Grafting influenced N content, with grafted plants having 12.2% larger values compared to non-grafted plants. Similarly, the P content in grafted plants was 15.9% higher than in non-grafted plants. The positive effects of grafting were greater for K^+^ and Mg^2+^ contents, exceeding by 26.7% and 20.3%, respectively, those observed in non-grafted plants. Regarding microelements, it was found that Mn^2+^ in grafted plants was 34.7% higher than in plants without grafting, the Zn^2+^ content was 19.5% higher in grafted plants, the Fe^2+^ content in grafted plants was 18.2% higher than plants without grafting. The Cu^2+^ content was also affected since grafted plants obtained 11.5% higher than plants without grafting.

### 2.2. Effect of ZnO NPs on Mineral Content

The use of nanoparticles favored the assimilation of mineral elements in bell pepper plants, reflecting a significant difference between them except for Ca^2+^ content. The N and P contents were affected, with the plants receiving ZnO NPs of 30 mg L^−1^ having 12.3% and 25.9% higher N and P values than the control plants. The Mg^2+^ content was 36.8% higher in the applied ZnO concentration of 30 mg L^−1^ compared to the control. The Mn^2+^ content was also affected using the 30 mg L^−1^ concentration since it increased by 42.2% compared to the control. The Zn^2+^ content was also affected using nanoparticles in the 30 mg L^−1^ concentration, showing increases of 27% over the control plants. The Fe^2+^ content increased by 45% with the concentration of 30 mg L^−1^ compared to the control. The Cu^2+^ content did not present significant differences between treatments (Table 1).

### 2.3. Effect of the Interaction between Grafting and ZnO NPs on Mineral Content

The combination of factors influenced the production variables (Table 1), where the highest results were obtained when using grafted plants and the application of ZnO NPs at a concentration of 30 mg L*^−^*^1^ for the variables N, Ca^2+^, Mg^2+^, and Mn^2+^. In plants with graft and without graft, no significant difference was found with the use of the concentration of 30 mg L*^−^*^1^, where the highest concentrations of P and Zn^2+^ were found. The K^+^ content did not show a significant difference between interactions of plants with graft and concentrations (0, 10, 20, 30 mg L*^−^*^1^) and plants without graft with a concentration of 30 mg L*^−^*^1^, but it did compare to control treatment in plants without graft. The highest Zn^2+^ content was found in the interaction of 30 mg L*^−^*^1^ and grafted plants, with a significant difference to non-grafted plants and 10 mg L*^−^*^1^ of nanoparticles. The highest concentration of Fe^2+^ was found in the interaction plants with graft and 30 mg L*^−^*^1^ with a significant difference. The Cu^2+^ content only showed a significant difference in the graft interaction and 30 mg L*^−^*^1^ compared to plants without grafting and 10 mg L*^−^*^1^.

### 2.4. Effect of Grafting on Nutraceutical Content

The antioxidant capacity in fruits was affected in grafted plants. It was increased by 2.44% compared to plants without grafting. The content of ascorbic acid in grafted plants was higher by 4.63% compared to plants without grafting. The total phenols were affected using grafts compared to plants without grafting, where there was an increase of 1.33%. The amount of glutathione was 7.18% higher in plants with a larger graft. Total proteins increased 1.08% in grafted plants compared to non-grafted plants (Table 2).

### 2.5. Effect of ZnO NPs on Nutraceutical Content

The nutraceutical quality improved with the dose of 30 mg L^−1^, and there was an upward trend with the statistical difference compared to the control. Antioxidant capacity increased 13.95%, ascorbic acid content increased 26.77%, total phenols increased 10.93%, glutathione content increased 11.46%, total protein content increased 11.01% (Table 2).

### 2.6. Effect of the Interaction between Grafting and ZnO NPs on Nutraceutical Content

The interaction of grafted plants and 30 mg L^−1^ of ZnO NPs resulted in significant increases in the nutraceutical content of antioxidant capacity, ascorbic acid, and total phenols. In glutathione content, there was no statistical difference in the interactions with grafted and non-grafted plants in concentrations of 10, 20, and 30 mg L^−1^, but with control treatments. The glutathione content did not present a statistical difference between the interactions of plants with graft, the control treatments, and 20 mg L^−1^ compared to plants without grafting and the treatments 10, 20, and 30 mg L^−1^. The interaction plants with graft and the control treatment did not present a statistical difference compared to the interactions of plants without graft and the treatments 10 and 20 mg L^−1^ (Table 2). The highest content of total proteins was found in plants with grafting and without grafting at a concentration of 30 mg L^−1^; on the other hand, the lowest contents were found in the control treatments (Table 2).

### 2.7. Effect of Grafting on Fruit Quality

Statistical difference was found in the fruit firmness variable since grafted plants were 8.8% firmer than plants without grafting. For fruit sugar content, no statistical difference was found between grafted and non-grafted plants (Table 3).

### 2.8. Effect of ZnO NPs on Fruit Quality

The firmness was statistically affected by the application of zinc oxide nanoparticles. The concentration of 30 mg L^−1^ had a difference compared to the control treatments, 10 and 20 mg L^−1^ obtaining a greater firmness exceeding by 17.7%, 10.2%, and 9.8%, respectively. The addition of ZnO NPs negatively affected the sugar content in the fruits since the control treatment showed a higher content that exceeded the plants treated with NPs by 34.6%, 24.3%, and 30.1% (Table 3).

### 2.9. Effect of Graft and ZnO NPs Interaction on Fruit Quality

The interaction between factors resulted in a statistical difference where the interaction of grafted plants and applications of 30 mg L^−1^ of ZnO NPs presented a difference compared to all other treatment interactions. Regarding the content of total soluble solids, the control treatments did not have a difference between control treatments in grafted and non-grafted plants but did in relation to the other treatment interactions (Table 3).

## 3. Discussion

### 3.1. Graft Effect on Mineral Content

Overall there was a trend of higher mineral contents (Table 1) in grafted plants. The higher N content observed in grafted plants agrees with those of Velasco-Alvarado et al. [32], who observed N contents being 46% higher than in grafted plants. Pilli et al. [33] also reported higher N contents in grafted tomato plants. Tagliavani et al. [34] argue that the concentrations of various nutrients are related to the vigor imparted by rootstock and scion combinations. For example, they mention that an increased content of N in grafted plants is associated with an abundant root system provided by the rootstock [35]. The ability to absorb and accumulate N by rootstocks depends on their efficiency to absorb and accumulate N [36]. The P content also increased with the use of grafts. Grafted plants have greater vigor, greater assimilation, and translocation of elements. Phosphorous is a mobile element in the plant and can be transported to other younger plant organs, as shown in a grapevine study by Gautier et al. [37]. Grapevine rootstocks altered the P concentrations of the stems, with the root system being the main route of P absorption [38].

The Ca^2+^ content in grafted plants did not present a statistical difference compared to plants without grafting, and it may be related to an inherent high or efficient ability of the hybrids used here for calcium absorption. These results agree with those reported by [39], where there were no significant differences between grafted and non-grafted tomato plants. For K^+^ contents, significantly higher values were also observed in grafted plants, similar to the work of Mayer et al. [40], who found a higher K^+^ content in grafted plums. The Mg^2+^ content also increased in our grafted tomato plants, agreeing with similar results obtained by Milenković et al. [41]. There was also a trend of higher content of the microelements Mn^2+^, Zn^2+^, Fe^2+^, and Cu^2+^ in grafted plants. The effect of grafting on a higher accumulation of minerals is due to the ability of the rootstock to facilitate greater absorption, transport of water and minerals through the roots [42]. Khan et al. [43] and others [44,45,46] have collectively reported that grafted plants are more efficient in their absorption of mineral nutrients (N, P, K^+^, Ca^2+^, Mg^2+^, Cu^2+^, Zn^2+^, and Mn^2+^), supporting the findings of the present research. The enhanced uptake efficiency might have also benefited from the use of an NFT system, which continuously provides essential nutrients and water to the root system [41,44,45].

### 3.2. Effect of ZnO NPs on Mineral Content

The macronutrients had a trend with the application of the 30 mg L^−1^ treatment (Table 1), where they presented the highest N concentrations, agreeing with similar results obtained by Gai et al. [46]. Dimkpa et al. [47] reported that maintaining an adequate water supply plus the application of ZnO NPs increased the general accumulation of N in sorghum by 8% and 16%, benefiting the translocation of nutrients. The content of P presented an increase with significance compared to the rest of the treatments, including the control. These results also agree with those obtained by Gai et al. [46], where the application of ZnO NPs increased the P content in stems and leaves in all treatments. As Zn is a structural component of phosphorous mobilizing phosphatase and phytase enzymes, application of ZnO NPs may help in more secretion of P-mobilizing enzymes, involved in native P mobilization for plant nutrition from unavailable organic sources [48]. The content of K^+^ in our pepper plants was statistically affected by a ZnO NPs concentration of 30 mg L^−1^, which had the highest K^+^ values compared to the control. These results agree with those obtained by Dimkpa et al. [49], who observed that Zn-containing micronutrient formulations enhanced K^+^ uptake in sorghum. The addition of ZnO NPs did not influence the Ca^2+^ content. Numerically, the highest content was found with the application of the 30 mg L^−1^ treatment, possibly influenced by the variability in the contents of the samples. The highest Mg^2+^ content was found in the treatment with the application of 30 mg L^−1^ of ZnO NPs. Dimkpa et al. [47] reported its Zn^2+^-promoting effect, benefiting the translocation and assimilation of Mg in the plant, increasing its absorption. Mg^2+^ participates in multiple functions including energy production, protein synthesis, cell membranes and chromosomes, ion transport across the cell membrane, and cell migration and interacts with protein, fiber, vitamin D, Ca^2+^, and Zn^2+^ [50].

The foliar application of ZnO NPs also affected micronutrient contents in the plants, with a trend of higher contents with the application of 30 mg L^−1^ ZnO NPs (Table 1). Bala et al. [51] report that when applying ZnO NPs, the Mn^2+^ content in rice grains increased 23.92% compared to the control. While there is no information on the effect of NPs on the Mn^2+^ content in other plants, it is known that Mn^2+^ is an essential microelement that participates in the normal plant development and growth and that an excess will produce intoxication problems [52].

Plant Zn^2+^ contents were also only significantly different with the application of 30 mg L^−1^ ZnO NPs compared to the control treatment. These results agree with those obtained by Rizwan et al. [53], where the application of ZnO NPs to seeds increased the concentrations of Zn^2+^ in the shoots by approximately 12% and 24% and in the roots by approximately 13% and 19% in relation to the controls. It is possible that nanoparticle treatments show higher foliar Zn^2+^ contents only because nanoparticulate residues left on the surface of leaves are difficult to be completely removed by washing protocols and not necessarily translocated into the tissues [54].

The 30 mg L^−1^ treatment of ZnO NPs enhanced tissue Fe^2+^ contents compared to the control treatment. The NRAMP3,−4 and MTP1,−8 are carriers of essential metals through vacuoles that regulate the translocation of Fe^2+^ and Cu^2+^, and their function is driven by Zn content sequestration in the tonoplast. The sequestration of Zn occurs through ZIP, NRAMP, and YSL transporters, and high levels of Zn may cause overexpression of essential minerals transporters, leading to a change of the efflux [55]. The ZnO NPs did not statistically affect the Cu^2+^ content in bell pepper plants. Previous studies have shown that ZnO NPs dissociate, releasing Zn ions into the medium [56], and according to Alloway [57], Zn decreases the uptake of Cu^2+^ because both metals are taken up by the same transporter. Faizan et al. [58] reported an increase in mineral content (N, K^+^, Zn^2+^, Mn^2+^, and Fe^2+^) due to the application of ZnO NPs.

### 3.3. Effect of the Interaction between Grafts and ZnO NPs on Mineral Content

Regarding the interaction of factors, there was a trend of higher content of N, P, Ca^2+^, and Mg^2+^ in the graft interaction and 30 mg L^−1^ of ZnO NPs (Table 1). These results may be due to the effect of ZnO NPs on minerals in plants depending on concentration and grafting, as reported by Salama et al. [59] who observed a significant increase in N, Fe^2+^, and Zn^2+^ contents in leaves and seeds at foliar applications of 40 ppm ZnO NPs. Regarding the macronutrient K^+^, the graft interaction and 10 mg L^−1^ obtained the highest concentration compared to the treatment without a control graft. In the same way, the content of micronutrients was found in greater quantity with the statistical difference in the graft interaction and the treatment of 30 mg L^−1^ of ZnO NPs, for the elements Mn^2+^, Zn^2+^, Fe^2+^, and Cu^2+^. One of the qualities of the rootstock in tomatoes is to prolong the absorption capacity of water and minerals [32]. Nanosized nutrients are more readily available to plant pores and are more efficient [60]. The role of Zn^2+^ as a cofactor for various enzymes and its particle size facilitates solubility and the ability to penetrate through the leaf surface and release Zn^2+^ ions through the cuticle [61].

### 3.4. Effect of the Graft on Nutraceutical Content

The nutraceutical content was affected by grafting and presented a trend of higher contents of antioxidant capacity, ascorbic acid, total phenols, glutathione, and total proteins (Table 2). García-López et al. [62] reported statistically higher contents of antioxidant capacity and total phenols in *Capsicum annuum* seedlings subjected to applications of 500 ppm ZnO NPs compared to the control. Djidonou et al. [63] reported a higher content of ascorbic acid in grafted tomato fruits compared to non-grafted control plants. The grafting factor helped to counteract stress in the plants, which increased the glutathione content for ROS capture compared to the control treatment. Zi-Kum et al. [64] reported a higher content of reduced glutathione in cucumber plants and fruits grafted under Cu^2+^ stress. Non-grafted plants had a higher concentration of ROS and a greater need for reduced glutathione to eliminate the effects of these radicals. Grafting increased the accumulation of total proteins. One of the main reasons is that grafted plants had a higher N content, which is a crucial and determining element in the protein content in the plant [65]. Sánchez-Torres et al. [66] reported higher nitrogen content, 17.8% and 16% total protein content in grafted pepper plants compared to non-grafted plants, which did not have the same capacity to absorb or assimilate nutrients as grafted plants. Altogether, the results obtained in this work point out that combining grafting with ZnO NPs applications will significantly enhance the absorption, transport, and accumulation of nutrients in bell pepper plants.

### 3.5. Effect of the ZnO NPs on Nutraceutical Content

The application of ZnO NPs statistically affected the nutraceutical variables evaluated: antioxidant capacity, ascorbic acid, total phenols, glutathione, and total proteins (Table 2), which presented the highest concentrations and a tendency to treatment 30 mg L^−1^ compared to control. Garcia-Lopez et al. [62] reported a significant accumulation of phenolic compounds in tomato radicles with the application of ZnO NPs at concentrations of 100 to 500 ppm, compared to the control. Similarly, the treatment of 500 ppm increased antioxidant capacity compared to the control. In addition, antioxidants could interfere with the oxidation process induced by various stresses acting as oxygen scavengers; therefore, the tolerance to stress by ZnO NPs might be correlated with an increase in the antioxidant potential [67].

Higher concentrations of total phenolic compounds were found in pomegranates exposed to Zn^2+^ + B_2_ treatments, while the lowest contents were present in fruits of the control treatment [68]. The ascorbic acid content in our bell peppers was also affected by the application of ZnO NPs, agreeing with Esfandiari et al. [69], whose zinc treatments increased the ascorbic acid content of wheat compared to their control. Plants have evolved various protective mechanisms to limit oxidative damage by the production of antioxidant non-enzymatic, phenols and ascorbic acid are highly correlated with the defense of the plants, it is possible that its content in the plant and fruit increases as a response to stress induced by ZnO NPs [70]. Generation of ROS and reactive nitrogen species and H_2_O_2_ upon exposure to ZnO engineered NP suggest that toxicity of ZnO NPs predominantly caused by both the particulates and ionic forms [71].

In other results reported by Tripathi et al. [72], ZnO NPs and SNP treatments stimulated the concentrations of glutathione species. Glutathione is related to stress response, and it is possible that protein increases as a response to NPs induced stress [73]. The application of ZnO NPs to plants significantly promoted the protein content compared to the control. These results agree with those reported by [74], the effect of ZnO NPs as a regulatory cofactor in protein synthesis in the plant [75]. The present results suggest that higher protein contents could protect cells from any oxidative stress caused by higher concentrations of NPs and higher total protein contents. Several proteins such as heat shock protein, catalase, and glutathione S-transferase are related to stress response [73], and those proteins were upregulated under abiotic or biotic stress. It is possible that protein increased as a response to NPs induced stress. Raliya et al. [76] Reported that nanoparticles were accumulated in roots, shoots, and leaves, independent of the application methodology. This indicates that once nanoparticles are absorbed by plants (either through root or leaf cells), they are bio-distributed throughout the plant by its vascular system.

### 3.6. Effect of the Interaction between Grafting and ZnO NPs on Nutraceutical Content

The interaction of grafting and ZnO NPs presented a trend of higher content of antioxidant capacity, ascorbic acid, total phenols, glutathione, and total proteins, all likely a result of greater plant vigor and assimilation and translocation of water and nutrients in grafted plants (Table 2) [62,64,65]. The effect of ZnO NPs generates associated changes in those of secondary compounds related to the defense of the plant against biotic and abiotic stress factors [77]. The secondary metabolites impact the nutritional quality of the edible parts of the plants as a stress-inducing factor that could increase the nutraceutical quality of the crops [78]. The effect of translocation, absorption of nutrients by the graft, and the effect of stress produced by the NPs could generate an increase in the nutraceutical quality in bell pepper fruits.

Increases in these important nutraceutical compounds are useful to protect plants from oxidative damage through their antioxidant activity [79]. Regarding glutathione content, grafted and non-grafted plants did not have statistical differences across applied concentrations of ZnO NPs.

### 3.7. Effect of Grafting on Fruit Quality

Fruit quality was affected by grafting, with grafted plants showing a higher firmness, likely influenced by their higher K^+^ content (Table 3). Botella et al. [80] reported that the K^+^ concentration affected the firmness of strawberry fruits, with lower concentrations decreasing firmness and higher K^+^ concentrations increasing it. Our results show that grafting improves the absorption of macro-minerals and microminerals, thus possibly benefiting the firmness of the fruit. Huang et al. [81] reported that rind thickness is significantly correlated with N concentration in the rind.

### 3.8. Effect of ZnO NPs on Fruit Quality

The application of ZnO NPs affected the quality of the fruits (Table 3), with 30 mg L^−1^ of ZnO NPs leading to higher firmness than other treatments. An apparent decrease in firmness could be due to the softening of the cell wall and the maturation activities related to enzymes such as pectin methyl esterase, where the activities of these enzymes could be delayed by ZnO NPs applications [82]. López-Herrera et al. [83] report that zinc applications in strawberry plants increased the firmness of the fruit compared to controls without Zn applications. The application of ZnO NPs negatively affected the sugar content in fruits compared to the control plants. These results may be related to increased respiration and metabolic activity in plants treated with ZnO NPs [84].

### 3.9. Interactive Effects of Grafting and ZnO NPs on Fruit Quality

The quality of the fruits, assessed by firmness, was significantly higher in grafted plants treated with 30 mg L^−1^ ZnO NPs, compared to all other treatments (Table 3). These results may be related to higher content of Ca^2+^ and K^+^ (Table 1), as these minerals enhance fruit firmness. Yfran et al. [85] reported that the firmness increased significantly in treatments with higher Ca^2+^ and K^+^ contents compared to controls, largely by promoting thicker fruit skins. Conversely, there were no statistical differences in the interaction of grafting with ZnO NPs applications, as they showed lower contents of soluble solids compared to the control plants. Faizan et al. [58] reported that the use of ZnO NPs is one of the new strategies to improve the growth and performance of plants under stress by trace metals.

## 4. Materials and Methods

The work was carried out in a greenhouse with passive climate control in the Department of Horticulture at Universidad Autónoma Agraria Antonio Narro, in Saltillo, Coahuila, México (latitude 25°21′23.4″, longitude 101°02′10.6″ and 1760 m above sea level).

### 4.1. Plant Material and Growing Conditions

The “SVEN RZ F1” hybrid (Rijk Zwaan) was used as the scion, which is a blocky type of pepper with short internodes, early, vigorous, and with great adaptability to greenhouse production and great fruit setting capacity in hot conditions, producing fine fruits. As rootstock, the hybrid “ULTRON F1” (HM CLAUSE) was used, which is a blocky type of pepper of indeterminate growth with great vigor, tolerant to salinity, and with yellow fruits.

The conditions inside the greenhouse were 4.5 W/m^2^ of solar radiation, a day maximum temperature of 36 °C and a minimum of 22 °C, and relative humidity of 40%. On February 7, 2020, the SVEN RZ F1 hybrid was sown in a 200-cavity polystyrene tray with peat as substrate. Ten days later, on February 17, 2020, the ULTRON F1 hybrid was sown in a 200-cavity tray, using peat as a substrate. The reason for sowing the rootstock 10 days later was due to its characteristic greater vigor and vegetative growth, thus allowing for a better match of the scion size and width of stems. This action allowed for equalization of stem diameters, benefiting the (graft) union of both plant structures (Figure 1a).

### 4.2. Graft

On March 7, 2020, the grafting was performed employing a splicing technique [86], performed when both structures had a stem diameter of two millimeters. The rootstock and the scion were cut at an angle of 60°, and both plant structures joined with a 2.0 mm diameter silicone clip. Once the grafting was carried out, the recovery of the plants began immediately, kept in a growing chamber with relative humidity conditions between 80% and 85% and a temperature of 25 to 28 °C. Ten days later, on March 17, 2020, the silicone clip was removed as a satisfactory union between scion and rootstock had occurred and healed. A mixture of water, 25% nutrient solution [87] and commercial foliar amino acids Metamin max^®^ (Agroestimulantes^®^ Mexicanos SA de CV., Aguascalientes, México) with a composition of 64.92% glutamic acid, 5.08% thiamine, and 30% inert conditioners, at a dose of 1 g L^−1^, were applied daily to the foliage with a manual sprinkler (Figure 1b).

### 4.3. Transplant to NFT System

Transplant to the NFT system was performed on April 13th and 14th, 2020. The seedlings were removed from the substrate, and their roots rinsed with water. Subsequently, the roots were immersed in a solution of 3% hydrogen peroxide as a preventive treatment against disease-causing microorganisms. Afterward, the plants were transferred to plastic baskets for hydroponics of 3 inches in diameter. For the adjustment of the plant inside the basket, a polyurethane sponge was used, leaving the root free so that it had contact with the nutrient solution (Figure 1c).

### 4.4. Synthesis and Characterization of ZnO NPs

The ZnO NPs used in the experiment were synthesized and characterized at Centro de Investigación en Química Aplicada (CIQA), in Saltillo, Coahuila, México (Figure 2). A controlled precipitation technique [88,89], employing a chemical hydrolysis method, was used as follows:
▪13.7 g of Zn(O_2_CCH_3_)_2_ and 600 mL of ethanol were placed in a ball flask with three necks;▪The solution was constantly stirred at 75 °C under reflux for 2 h;▪Then an aqueous solution of 0.22 M NaOH and an additional 100 mL of distilled H_2_O were added to complete the reaction mixture;▪It was stirred for 24 h;▪The obtained ZnO NPs were immersed in ethanol and recovered by centrifugation (15,000 rpm/5 min);▪The precipitate was washed two times with ethanol and dried in an oven at 60 °C for 24 h;▪The dried ZnO NPs were crushed in an agate mortar to obtain a fine powder;▪Size and morphology of nanoparticles were measured by means of a high-resolution transmission electron microscope (HRTEM) Titan 80–300 kV (FEI Company, Hillsboro, OR, USA).

Concentrations of ZnO NPs were prepared at 10, 20, and 30 mg L^−1^, plus a control. Each solution was placed in a manual spray bottle with a capacity of 1 L and was applied homogeneously, on the upper and lower sides of the leaves, on the stems, flowers, and fruits (aerial parts). A volume of 250 mL per plant was applied, with droplets of 0.5 mm size. As previously reported, ZnO NPs applied as a foliar spray can be efficiently transported in the plant system [90].

The first application of ZnO NPs was made 10 days after grafting, on March 27th, 2020. The second application of ZnO NPs was made at the flowering stage, 80 days after transplant. The third application was carried out in the fruit filling stage, 95 days after transplant, in plants already established in the NFT system. The applications were made in the afternoon when temperatures averaged 22 °C and radiation 4.0 W/m^2^ day to avoid any problems from stressful temperatures and radiation (Figure 3).

### 4.5. Mineral Analysis

#### 4.5.1. Nitrogen Determination

For this variable, the micro Kjeldahl method was used. Four plants were taken randomly per treatment during the production stage, including leaves, stems, fruits, and roots. For the digestion stage, 0.1 g of sample, 1 g of catalyst mixture, and 3 mL of sulfuric acid were added to a 100 mL micro Kjeldahl flask. The flasks were placed in a LABCONCO electronic digester for 45 min until reaching a mint green color. Distillation stage: in an 800 mL round bottom flask, the digestion, 100 mL of distilled water, 30 mL of boric acid, 20 mL of sodium hydroxide, and 6 drops of indicator were added. The collected sample was taken to the distiller, where 125 mL of distillate (with a blue hue) was collected in a 250 mL Erlenmeyer flask. Titration stage: in this stage, the sample collected from the distillation was titrated with 0.01 N hydrochloric acid until it reached a reddish hue. The titrator was measured and recorded for the determination [91].

#### 4.5.2. Determination of N, P, K^+^, Ca^2+^, Mg^2+^, Mn^2+^, Cu^2+^, Fe^2+^, and Zn^2+^

0.1 g of previously ground and sieved sample were weighed and placed in porcelain crucibles and incinerated in an electronic muffle at 550 °C for 5 h. A total of 1 mL of distilled water, and 1 mL of concentrated hydrochloric acid were added to the incinerated samples. They were placed on a thermal plate at 200 °C for 30 min, allowed to cool to room temperature using 25 mL of hydrochloric acid. The samples were filtered with a Whatman No. 1 filter paper and stored in plastic bottles. The determination of phosphorus was carried out with an optical spectrophotometry technique [91]. For the determination of Ca^2+^, Mg^2+^, K^+^, Mn^2+^, Cu^2+^, Fe^2+^, and Zn^2+^, the filtrate from the aforementioned incineration was used, and it was determined by an atomic absorption spectroscopy technique [91].

The results of N, P, K^+^, Ca^2+^, Mg^2+^, Mn^2+^, Zn^2+^, Fe^2+^, and Cu^2+^ were expressed in milligrams per gram of sample (mg g^−1^). Standards of each mineral were used as the basis for calibration curves.

### 4.6. Biochemical Analysis

Five peppers were harvested representative of each plant with commercial maturity per eight treatments, making up a total of 40 samples cut into small squares with a knife. Each sample consisted of 30 g of cut pepper, stored in plastic jars, and kept in an ultra-freezer at −80 °C for 48 h. Thereafter they were freeze-dried at −80 °C with a vacuum pressure of 0.020 mbar for 72 h. The samples were extracted and macerated with a porcelain mortar and stored in plastic jars in a cool, dark place.

The extraction was carried out with 100 mg of lyophilized and macerated tissue that was placed in 2 mL microtubes, then 2.0 mL of the 1:1 water:acetone mixture was added, and it was vortexed for 30 s. Subsequently, the sample was sonicated for 5 min, centrifuged at 12,000 rpm for 10 min at 4 °C. The supernatant was extracted with plastic syringes, filtered on 0.45-micron pore filters, and stored in microtubes.

#### 4.6.1. Antioxidant Capacity

It was determined with the radical DPPH (2,2-diphenyl-1-picrylhydrazyl) using the commercial antioxidant test kit (Sigma-Aldrich^®^ CS0790, St. Louis, MI, USA). For the measurement of absorbances, a UV-Vis spectrophotometer at a wavelength of 530 nm was used. The antioxidant activity was determined by plotting the absorbances along a standard Trolox concentration curve and was expressed as the antioxidant capacity of DPPH (mmol Trolox EQ 100 g dry weight).

#### 4.6.2. Ascorbic Acid

The quantification was carried out in a Varian^®^ high-performance liquid chromatography HPLC (SpectralLab Scientific Inc. Markham, ON, Canada) under the following conditions: wavelength 230 nm, mobile phase NaH_2_PO_4_ 50 mM pH 2.8/acetonitrile in a ratio 80:20, flow of 1.0 mL/min. The column used was an aquasil C-18 at a temperature of 60 °C. The ascorbic acid concentration was reported in equivalent grams of Trolox per kilogram of sample, calculated from the Trolox and DPPH calibration curve. [92].

#### 4.6.3. Total Phenols

They were quantified using the Folin–Ciocalteu reagent, as described in [93]. The concentration of phenols was reported in equivalent grams of gallic acid per kilogram of sample (mg g^−1^ dry weight).

#### 4.6.4. Reduced Glutathione (GSH)

Glutathione activity was determined by the technique of [94,95]. A total of 480 µL of the biomolecule extract and 2.2 mL of 0.32 M Na_2_HPO_4_ were placed in a test tube, shaken, 320 µL of 1 mM DTNB were added, and the mixture was shaken again. The readings were taken with a UV-VIS spectrophotometer (SpectralLab Scientific Inc. Markham, ON, Canada) at 412 nm, the glutathione (GSH) content was expressed (mmol 100 g^−1^ dry weight).

#### 4.6.5. Total Protein

The determination was carried out using the colorimetric technique of Bradford [96], a spectrophotometer was used at a wavelength of 595 nm, the protein content is expressed as grams per kg of dry weight of plant material (g kg^−1^ dry weight).

#### 4.6.6. Fruits Firmness

For the determination of the firmness variable in bell pepper fruits, the following methodology was used, the fruit was placed horizontally and by compression using a conical strut in a force meter (QA SUPPLIES penetrometer) with a penetration of 2 mm. The results were reported as Kg force (Figure 4).

#### 4.6.7. Total Soluble Solids

To analyze the internal characteristics of the fruits, small squares of fruit were cut with a razor. In a mortar, fresh fruit was macerated, obtaining an approximate 5 mL of juice, a drop (0.5 mL) of fruit pulp was placed in the digital refractometer HANNA model HI 96801 (HANNA Instruments Inc, Washington, USA), and the reading was expressed as a percentage of concentration of total soluble solids as °Brix (Figure 5).

The experimental design used for this experiment was completely randomized with a factorial arrangement (2 × 4). With the data obtained, an analysis of variance (ANOVA) was made, and for the detection of statistical differences between treatments, Tukey’s mean comparison test (α = 0.05) was used. The factors were with and without graft, and 4 concentrations of ZnO NPs (0, 10, 20, 30 mg L^−1^), resulting in 8 treatments and 4 repetitions. InfoStat Statistical software was used to analyze the information (InfoStat version 1.0).

## 5. Conclusions

The results from the present experiment support the hypothesis that the combined use of grafting and ZnO NPs increased the nutritional quality of bell pepper fruit. Furthermore, an increase in the levels of nutraceutical components and overall fruit quality was also observed. These results support the recommended use of grafting and foliar ZnO NPs applications as cultural crop management tools that could significantly improve the quality and nutritional content in the bell pepper crops, possibly giving greater benefits to the health of the consumers. We recommend that in future experiments, higher concentrations of ZnO NPs be used, as well as evaluating their application and responses in other horticultural crops.

## Figures and Tables

**Figure 1 plants-10-02793-f001:**
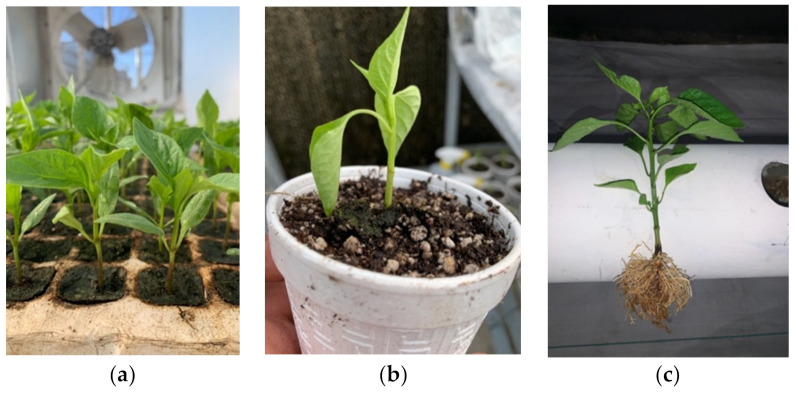
Treatments application: (**a**) sowing and management of scions and rootstocks; (**b**) plant 10 days after the graft; (**c**) root wash and transplant to NFT system.

**Figure 2 plants-10-02793-f002:**
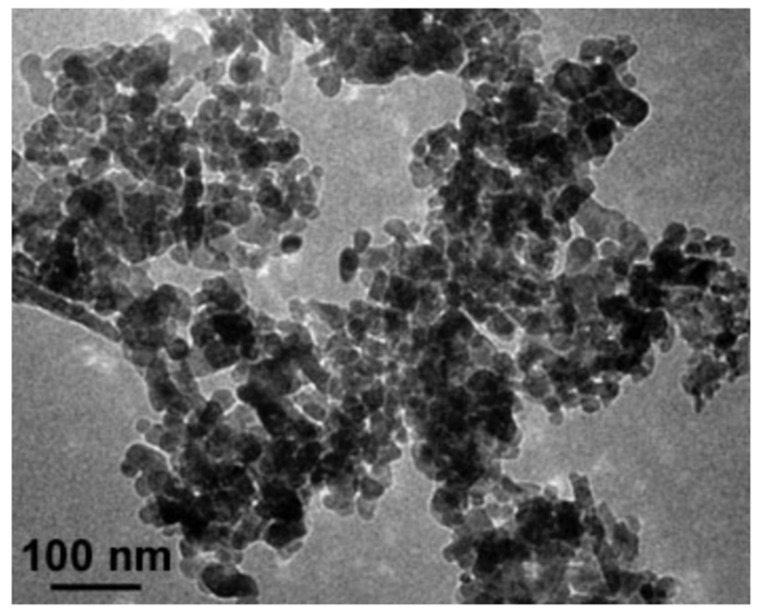
Photograph of zinc oxide nanoparticles.

**Figure 3 plants-10-02793-f003:**
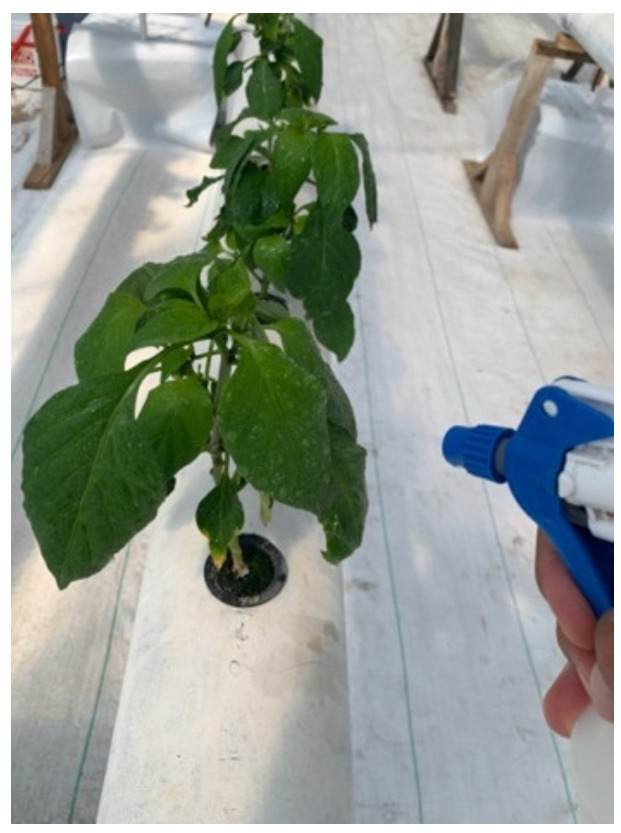
Foliar application of zinc oxide nanoparticles.

**Figure 4 plants-10-02793-f004:**
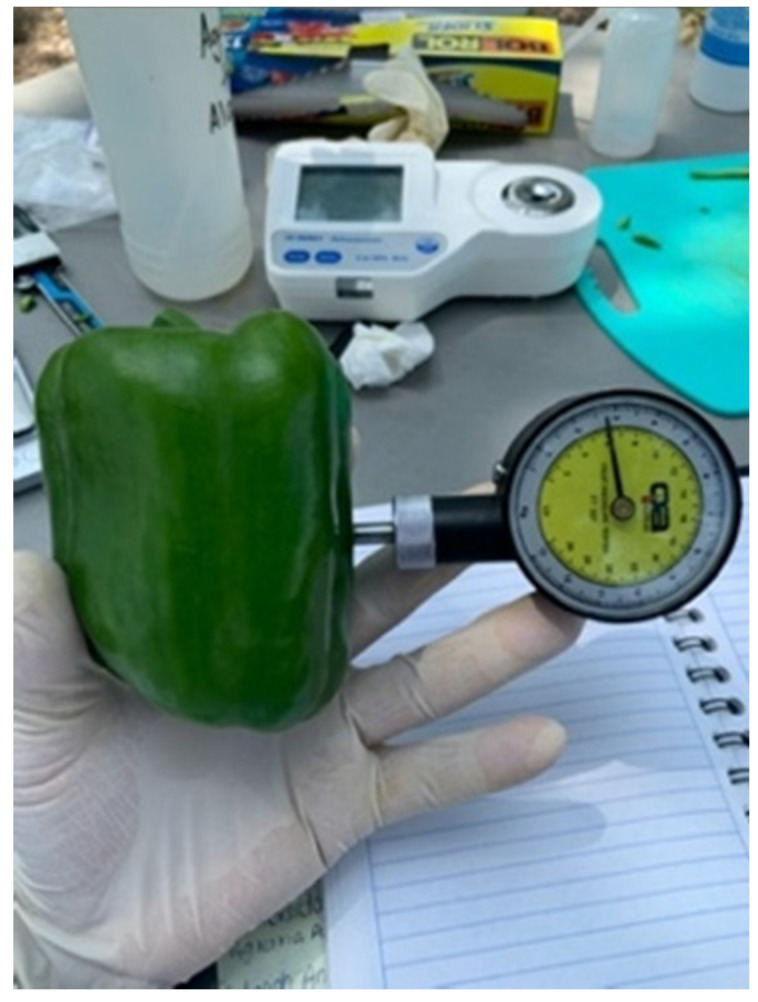
Use of manual penetrometer in bell pepper fruit.

**Figure 5 plants-10-02793-f005:**
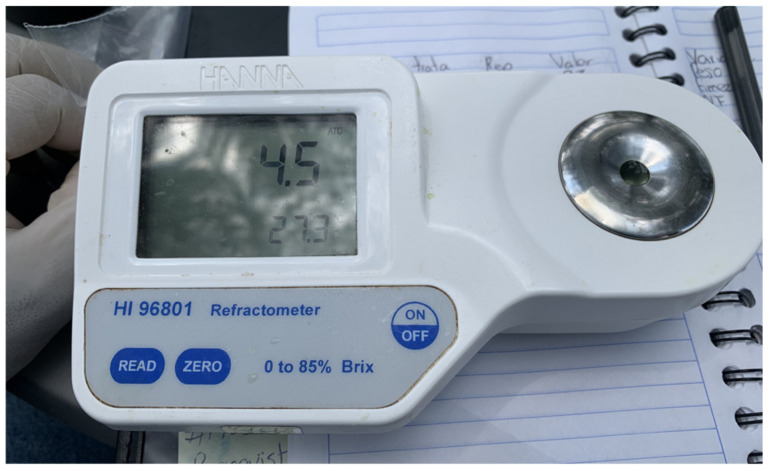
Use of digital refractometer for measurement of total soluble solids 4.8. Experimental design and statistical analysis.

**Table 1 plants-10-02793-t001:** Comparison of means of graft effects, ZnO NPs, and interaction in mineral content.

Factor		N mg g^−1^	P mg g^−1^	K^+^ mg g^−1^	Ca^2+^ mg g^−1^	Mg^2+^ mg g^−1^	Mn^2+^ mg g^−1^	Zn^2+^ mg g^−1^	Fe^2+^ mg g^−1^	Cu^2+^ mg g^−1^
Grafted		36.2 a	12.6 a	40.4 a	13.8 a	7.4 a	0.033 a	0.092 a	0.050 a	0.008 a
Ungrafted		31.8 a	10.6 b	29.6 b	11.3 a	5.9 b	0.021 b	0.074 b	0.041 b	0.007 b
Control		32.8 bc	10.6 b	28.9 b	12.1 a	5.5 c	0.021 c	0.073 b	0.032 c	0.009 a
10 mg L^−1^		32.2 c	10.4 b	37.6 ab	10.8 a	5.5 c	0.022 c	0.082 b	0.040 bc	0.008 ab
20 mg L^−1^		33.7 b	11.1 b	33.1 ab	11 a	6.9 b	0.028 b	0.077 b	0.049 ab	0.008 ab
30 mg L^−1^		37.4 a	14.3 a	40.4 a	16.3 a	8.7 a	0.037 a	0.100 a	0.059 a	0.009 a
Grafted	Control	35.6 b	11.3 bc	36.5 abc	11.7 b	5.8 bc	0.025 bc	0.084 bc	0.040 abc	0.009 a
	10 mg L^−1^	34.5 b	10.8 c	52.2 a	8.7 b	5.4 c	0.027 bc	0.094 ab	0.050 ab	0.008 ab
	20 mg L^−1^	36 b	12.3 b	35.3 abc	12.1 b	7.3 b	0.032 b	0.081 bc	0.051 ab	0.008 ab
	30 mg L^−1^	38.6 a	16 a	36.7 abc	22.7 a	11 a	0.047 a	0.109 a	0.059 a	0.009 a
Ungrafted	Control	29.9 c	10 c	21.2 c	12.5 b	5.3 c	0.017 c	0.090 ab	0.025 c	0.007 ab
	10 mg L^−1^	29.8 c	10.8 c	23 c	12.9 b	5.7 c	0.017 c	0.070 bc	0.031 bc	0.007 b
	20 mg L^−1^	31.3 c	10 bc	30 bc	9.9 b	6.4 bc	0.024 bc	0.073 bc	0.047 ab	0.007 ab
	30 mg L^−1^	36.2 b	12.5 bc	44.2 ab	10 b	6.4 bc	0.027 bc	0.090 ab	0.060 a	0.008 ab
Grafted		***	**	**	ns	***	***	***	*	***
ZnO NPs		***	***	*	ns	***	***	**	***	ns
Graft x NPs		*	ns	**	**	***	*	ns	ns	ns
CV %		2.96	13.32	21.77	33.10	10.45	15.32	12.91	20.29	7.19
SD		0.32	0.24	1.19	0.55	0.19	8.83	10.96	14.45	0.80

Note. Abbreviations: nitrogen content (N), phosphorus content (P), calcium content (Ca^2+^), potassium content (K^+^), magnesium content (Mg^2+^), manganese content (Mn^2+^), zinc content (Zn^2+^), iron content (Fe^2+^), copper content (Cu^2+^), means with different letters are significantly different, the test of comparison of means by Tukey. ns = (*p* > 0.05), * = (*p* ≤ 0.05), ** = (*p* ≤ 0.01), *** = (*p* ≤ 0.001), CV = coefficient of variance.

**Table 2 plants-10-02793-t002:** Comparison of means of the graft effects, ZnO NPs, and interaction in the nutraceutical content.

Factor		Antioxidant Capacity (µMTrolox EQ 100/g DW)	Ascorbic Acid (g/kg DW)	Total Phenols (g/kg DW)	Reduced Glutathione (mmol 100 g ^−1^ DW)	Total Protein (g/kg DW)
Grafted		0.82 a	1033.55 a	9.77 a	7.24 a	56.64 a
Ungrafted		0.80 b	985.70 b	9.64 b	6.72 b	55.93 b
Control		0.74 d	848.70 d	9.05 d	6.41 b	52.92 d
10 mg L^−1^		0.80 c	950.69 c	9.74 c	6.97 a	55.32 c
20 mg L^−1^		0.82 b	1080.21 b	9.88 b	6.89 a	57.43 b
30 mg L^−1^		0.86 a	1158.90 a	10.16 a	7.24 a	59.47 a
Grafted	Control	0.76 d	864.28 f	9.12 e	6.51 bc	53.42 e
	10 mg L^−1^	0.80 c	984.21 d	9.79 cd	7.22 a	55.49 d
	20 mg L^−1^	0.83 bc	1103.50 b	9.92 bc	7.07 ab	57.79 bc
	30 mg L^−1^	0.88 a	1182.19 a	10.27 a	7.32 a	59.86 a
Ungrafted	Control	0.73 d	833.12 f	8.99 e	6.30 c	52.42 e
	10 mg L^−1^	0.80 c	917.16 e	9.70 d	6.72 abc	55.15 d
	20 mg L^−1^	0.82 bc	1056.92 c	9.84 cd	6.70 abc	57.07 c
	30 mg L^−1^	0.84 b	1135.61 b	10.04 b	7.16 ab	59.09 ab
Grafted		**	***	***	*	**
ZnO Nps		***	***	***	***	***
Graft *x* NPs		*	*	*	ns	ns
CV %		1.99	1.77	0.74	4.71	1.27
SD		0.05	124.14	0.42	0.46	2.58

Means with different letters are significantly different, the test of comparison of means by Tukey. ns = (*p* > 0.05), * = (*p* ≤ 0.05), ** = (*p* ≤ 0.01), *** = (*p* ≤ 0.001), CV = coefficient of variance.

**Table 3 plants-10-02793-t003:** Comparison of means of graft effects, ZnO NPs, and interaction on fruit quality.

Factor		Fruits Firmness (Kg)	Sugar Content (°Brix)
Grafted		2.38 a	5.26 a
Ungrafted		2.17 b	5.11 a
Control		2.09 b	6.67 a
10 mg L^−1^		2.29 b	4.66 bc
20 mg L^−1^		2.18 b	5.05 b
30 mg L^−1^		2.54 a	4.36 c
Grafted	Control	2.10 b	6.84 a
	10 mg L^−1^	2.34 b	4.78 bc
	20 mg L^−1^	2.20 b	4.80 bc
	30 mg L^−1^	2.88 a	4.22 c
Ungrafted	Control	2.08 b	6.50 a
	10 mg L^−1^	2.24 b	4.54 bc
	20 mg L^−1^	2.16 b	5.30 b
	30 mg L^−1^	2.20 b	4.10 c
Grafted		**	ns
ZnO NPs		***	***
Graft × NPs		**	ns
CV %		8.69	9.64
SD		0.30	1.03

Means with different letters are significantly different, the test of comparison of means by Tukey. ns = (*p* > 0.05), ** = (*p* ≤ 0.01), *** = (*p* ≤ 0.001), CV = coefficient of variance.

## Data Availability

Not applicable.

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
