# Peer review of "Effect of Graft and Nano ZnO on Nutraceutical and Mineral Content in Bell Pepper"

_plants, 2021, doi:10.3390/plants10122793_

Round 1

Reviewer 1 Report

Title of the article may be change to "Effect of graft and nano ZnO on nutraceutical and mineral content in bell pepper" 

Authors observed that the content of N,  P, K+ , Ca2+, Mg2+, Mn2+, Zn2+, Fe2+, Cu2+, total antioxidants, ascorbic acid, total phenols, glutathione, 22 total proteins, fruit firmness were improved with the graft and the use of nanoparticles but - why these improvement happened, need to explain the mechanism.

Author need to discuss the results in detail with the existing literature. 

Author should discuss the results of nanoparticle characterization. 

Cite the following paper from mechanism view point - 

Raliya, Ramesh, and Jagadish Chandra Tarafdar. "ZnO nanoparticle biosynthesis and its effect on phosphorous-mobilizing enzyme secretion and gum contents in Clusterbean (Cyamopsis tetragonoloba L.)." Agricultural Research 2.1 (2013): 48-57.

Raliya, R., Nair, R., Chavalmane, S., Wang, W. N., & Biswas, P. (2015). Mechanistic evaluation of translocation and physiological impact of titanium dioxide and zinc oxide nanoparticles on the tomato (Solanum lycopersicum L.) plant. Metallomics7(12), 1584-1594.

Nanofertilizer for precision and sustainable agriculture: current state and future perspectives. Journal of Agricultural and Food Chemistry66(26), 6487-6503.

Author Response

Consulte el archivo adjunto.

Reviewer 2 Report

The manuscript evaluates the effects produced by graft and zinc oxide nanoparticles on the nutritional composition of the bell pepper. Introduction and Discussion sections need to be improved. Specific comments, by line(s):

Change in Line 18 and throughout, “zinc oxide nanoparticles (NPs ZnO)” to “zinc oxide nanoparticles (ZnO NPs)”

Change in Line 21 and throughout “NPs ZnO” to “ZnO NPs”

Change in Line 26 “increased the content of:” to “increased the: content of”.

Reconsider the Keywords used and possible increase their number.

In Introduction you can add a paragraph related to the importance of Zn in human diet and the toxicities of Zn deficiency in plant growth and development. Zn fortification is important in improving nutrient utilization efficiency and plant growth (Agriculture 2021, 11, 505). Zn deficiency is leading to reduced crop yield and poor nutritional quality of grains and derivatives (Science of the Total Environment 2020, 738, 140240). Foliar sprayed Zn NPs have been proposed as nonphytotoxic fungicides as well as fertilizers (ACS Appl. Mater. Interfaces 2018, 10, 4450−4461). Relevant discussion on the importance of Zn in human nutrition and the benefits of ZnO NPs as fertilizers can be included in Discussion section.

In table 1 some results are given as % and some others as mg/Kg. Chang all results as mg gr-1 and provide SE or SD in all of them.

Change also mg/L to mg L-1.

Change in line 282 “agree with those reported by [53].” to “agree with those reported previously [53].”

Delete lines 335-337 and check carefully the whole manuscript.

Figures must be placed near their first mention and must have a more detail legend.

Reviewer 3 Report

NUTRACEUTICAL AND MINERAL CONCENTRATION IN 2 BELL PEPPER FRUITS GRAFTED AND GROWN WITH ZnO NPs

General Comments

Very well collated manuscript. The introduction clearly explains all the aspects of the study background. The results and discussions have been smoothly explained and are easy to comprehend. The manuscript is technically sound and possesses a good presentation. A few basic comments regarding the manuscript can be listed as under:

Major:

  • The author is suggested to highlight the replicates of reading taken for biochemical analysis
  • The author is suggested to present data in graphical plots. Though the tables presented in the manuscript have all needful statistical analysis done graphical illustration would be better with error bars
  • Kindly remove the effect from figure 1 and keep it simple with a fine-line border or without a border
  • Authors may consider this recently published MDPI article: DOI: 10.3390/plants10112254

Minor:

  • The units, mgL-1 and gL-1 are inconsistent; mg/L and g/L have also been used. Please use consistent units in the entire manuscript
  • Ln 436: ‘5 peppers five fruits’; please clarify the statement
  • Minor punctuation and grammatical errors in the manuscript should be corrected
  • Inconsistent abbreviation in the manuscript: zinc oxide nanoparticle (ZnO NP)
  • Abstract: Ln 19: zinc oxide nanoparticle is abbreviated as NP ZnO; instead of ZnO NP
  • Abstract: Ln 20: ‘consistently with’; instead of ‘consistent of’
  • Keywords have been numbered inappropriately
  • Ln 43: Full form of NFT has not been elaborated in the first occurrence in the manuscript
  • Ln 335: preposition error: ‘divided into’; instead of ‘divided by’
  • Please check reference number 66 in the bibliography.

Reviewer 4 Report

Interesting article on a very important aspect of pepper cultivation.
An article prepared correctly with a well-presented purpose.
Correctly selected methodologies in addition to the method of liquid application with zinc nanoparticles. Manual spraying is never too accurate and repeatable. There is also no information about the amount of liquid applied to each plant and the size of the drops (fine, medium, large). In the future, I suggest spraying with a portable battery sprayer, measuring the spraying time of each plant and providing the type of spraying nozzle and pressure.
For the results in the tables, please provide the standard deviation (SD)

Author Response

Sorry for the delay, but I couldn't find the analysis of variance.

Round 2

Reviewer 2 Report

In the final corrected version of the manuscript some corrections made in the responce to the reviewer's file are not included. For example: Lines 17-18 and throughout.

When you write zinc oxide nanoparticles and you want to abbreviate it you must keep the same order in the abbreviation as in the text so, zinc oxide nanoparticles is abbreviated ZnO NPs.
